# Three Dimensional Models of Endocrine Organs and Target Tissues Regulated by the Endocrine System

**DOI:** 10.3390/cancers15184601

**Published:** 2023-09-17

**Authors:** Edlira Luca, Kathrin Zitzmann, Stefan Bornstein, Patrick Kugelmeier, Felix Beuschlein, Svenja Nölting, Constanze Hantel

**Affiliations:** 1Department of Endocrinology, Diabetology and Clinical Nutrition, University Hospital Zurich (USZ) and University of Zurich (UZH), 8091 Zurich, Switzerland; 2Department of Medicine IV, University Hospital, LMU Munich, 80336 München, Germany; 3Medizinische Klinik und Poliklinik III, University Hospital Carl Gustav Carus Dresden, 01307 Dresden, Germany; 4Kugelmeiers AG, 8703 Erlenbach, Switzerland; 5Endocrine Research Unit, Medizinische Klinik und Poliklinik IV, Klinikum der Universität München, 80336 Munich, Germany

**Keywords:** endocrine organs, neuroendocrine neoplasia, spheroids, organoids, pituitary, thyroid, adrenal, pancreas, ovaries, breast, prostate

## Abstract

**Simple Summary:**

The intricate endocrine system regulates human physiology by producing and secreting various hormones. Novel methodology pertaining to spheroids and organoids is currently used to understand endocrine gland development and function, as well as to identify novel therapeutic agents to target endocrine cancers. Cells cultured in non-adherent conditions aggregate to form 3-dimentional spheres. Spheroids results from aggregates of one cell type, while multiple cells types self-organize together into organoids. Organoids are multifaceted structures that capture the intricacies of cell-cell interactions and mimic the biological complexity of organs in a dish. Additionally, patient tumor biopsies cultured as organoids can recapitulate the phenotype and genotype of the original tumors in vitro, thereby advancing therapeutic strategies for personalized medicine. In this manuscript, we will review the efforts to implement spheroids/organoids technology in basic research to answer fundamental questions in endocrine gland physiology as well as in preclinical research for cancers of endocrine organs and some of their target tissues.

**Abstract:**

Immortalized cell lines originating from tumors and cultured in monolayers in vitro display consistent behavior and response, and generate reproducible results across laboratories. However, for certain endpoints, these cell lines behave quite differently from the original solid tumors. Thereby, the homogeneity of immortalized cell lines and two-dimensionality of monolayer cultures deters from the development of new therapies and translatability of results to the more complex situation in vivo. Organoids originating from tissue biopsies and spheroids from cell lines mimic the heterogeneous and multidimensional characteristics of tumor cells in 3D structures in vitro. Thus, they have the advantage of recapitulating the more complex tissue architecture of solid tumors. In this review, we discuss recent efforts in basic and preclinical cancer research to establish methods to generate organoids/spheroids and living biobanks from endocrine tissues and target organs under endocrine control while striving to achieve solutions in personalized medicine.

## 1. Introduction

The endocrine system produces and secretes hormones to regulate the functions of human physiology (Figure 1) [1]. Endocrine cells, located within specialized glands or dispersed throughout nonendocrine organs, include steroid-producing cells (adrenal cortex and ovaries), thyroid follicular cells (synthesize thyroid hormones and thyroglobulin) and neuroendocrine cells. Neuroendocrine cells are a distinct part of the endocrine system, located diffusely in endocrine glands as well as in nonendocrine organs and subdivided into epithelial cells derived from the endoderm and nonepithelial paraganglia derived from the neuroectoderm. Endocrine cancers include those in the pituitary, thyroid, parathyroid, adrenal, testis, ovaries and neuroendocrine tumors. Dysregulation of hormonal systems not only results in pathological abnormalities in the affected gland locally, but also in manifestations in endocrine-sensitive organs due to excess of circulating hormones.

Over the past two decades, preclinical research has turned increasingly more to spheroids and organoids to investigate tissue pathophysiology and responses to current and novel cancer therapies. Organoids are heterogeneous self-organizing 3D aggregates that can recapitulate the structure, function and thereby overall biological complexity of organs [2]. Organoids can be regarded as miniorgans in a dish, as those arising from the differentiation of pluripotent stem cells, which allow for a greater in-depth analysis of organogenesis and organ function [3,4]. Additionally, organoids can be generated from patient tumor samples where the various cell types are allowed to conglomerate in vitro and recreate the tumor microenvironment [5]. Spheroids are similar but typically arise from the aggregation of one cell type, such as immortalized cell lines. Therefore, organoid and spheroid cultures can benefit a wide range of applications, from answering basic biology questions, to aiding in drug discovery and driving personalized medicine. Additionally, disorders of the endocrine system are unique in that treatment often necessitates hormone replacement therapies, which cause unwanted side effects and do not restore full organ function. Therefore, especially for endocrine organs, organoids offer the possibility of human transplantation and organ replacement, as is being developed for pancreatic islets in the case of type I diabetes [6,7] and for thyroid resections [8,9,10]. In this review, we emphasize how the implementation of in vitro organoid/spheroid cultures has expanded our basic understanding of endocrine tissue function and influenced preclinical research in endocrine tumors.

## 2. Advantages of Spheroids/Organoids

Primary human cells and immortalized cell lines have contributed valuable information to our basic understanding of tissue function as well as to cancer biology [5,11,12]. Two dimensional cultures are easy to grow, easy to use and efficient to analyze by functional assays and microscopy. However, in vivo, cells receive a multitude of cues due to the chemical and mechanical properties of the extracellular matrix and participate in complex juxtacrine, paracrine and cell–cell signaling due to architectural organization. Since these conditions are difficult to replicate in vitro in monolayers, preclinical research has turned to engineering 3D cultures where cells are grown as spheroids or organoids to provide more reliable biological information and improve drug discovery. Spheroids/organoids are generated in suspension, under nonadherent condition, with or without the help of matrixes.

There are similarities and differences between spheroids and organoids. In both cases, cells use adhesion molecules such as integrins and cadherins to interact with their environment and with other cells to self-assemble into aggregates in vitro [13,14]. Spheroids are typically generated from cell lines or primary cells and exhibit low structural complexity. Spheroid formation is achieved in less than one week in culture medium supplemented with FBS and often does not require addition of growth factors [14]. Spheroids from cancer cell lines are typically more resistant to therapeutic agents than the same cells grown as monolayers [15,16]. Organoids are composed of several cell types, can display intricate morphology and recapitulate organ structure and function. The starting material for organoids can be embryonic stem cells, which are differentiated into miniorgans in stepwise fashion through exposure to different growth factors. Additionally, organoids can be generated from normal adult tissue or patient tumors, as discussed below. In general, organoids are grown in scaffolds such as hydrogels of Matrigel and require growth factors for proper growth and expansion. To successfully achieve organoids from normal tissue, the starting material must be collected fresh in cooling conditions, gently dissociated and plated directly in settings that favor self-assembly of organoids [17]. There are also drawbacks to using spheroids/organoids. Most cellular assays are intended for monolayer cultures and one must ensure that reagents sufficiently penetrate spheroids/organoids. Another drawback is genetic shift in organoids cultured over multiple passages. Thirdly, there is a need to further develop, standardize and optimize organoid cultures, for instance, growth factors, media components, and the absence/presence/type of matrix used to grow PDOs. These reagents can influence the sensitivity and resistance of PDOs to therapeutic drugs, rendering the findings difficult to extrapolate to treatment options for the patient [18].

For cancer research, spheroids/organoids of uniform shape and size are highly desirable for quality and reproducibility of the data. Shape and size can be regulated by different plating techniques, as discussed below. The multicellular arrangement and large size of the spheroids/organoids advantageously allow for the formation of hypoxic and pathophysiological conditions within these cell masses. In vivo, hypoxic regions within tumors drive resistance to radio and chemotherapy [19,20,21]. The hypoxic environment reprograms cellular metabolism and affects proliferation, migration, invasiveness, and positively correlates with degree of malignancy and resistance to treatment. The architecture of large spheroids/organoids mimics the in vivo condition, whereby metabolically active and proliferating cells are located on the outside, followed by a zone of quiescent cells in the inside and a necrotic core. This characteristic organization results from the limit of diffusion of oxygen and nutrients (in the range 150–200 um) [13,14,22,23,24]. Appearance of hypoxic cores is well known for growing solid tumors. Interestingly, recent data suggest that these conditions might be relevant drivers of metastasis [25], and it is yet to be determined if resulting metastasis maintain the same necrotic properties after dissemination. Therefore, organoids and spheroids can serve as promising tools for such investigations.

Many different platforms exist to generate spheroids/organoids, from embedding in Matrigel [26], ultra-low attachment plates [27], hanging drop plates [28], microfluidic options and high-throughput devices such as microwell plates (Figure 2) [17]. In the case of matrixes, a uniform suspension is formed with Matrigel, for example, and then plated in normal culture plates such that the arising 3D structures are directly embedded into the matrix [26]. High-throughput drug screens, however, necessitate vast amounts of organoids to be generated and screened in a short length of time. Microwell plates represent the ideal platform to control the size and shape of organoids, which can be regulated via the initial amount of cells plated, time spent in culture, and proliferation rates of the cells. Ultra-low attachment (ULA) noncoated microwell plates, available in 96-well and 384-well format, display low adhesion properties, thereby forcing intracellular interactions and spheroid/organoid formation. Our group has worked extensively with the high-performing Sphericalplate 5D , a modified specially coated plate with 9120 microwells, to generate spheroids/organoids of uniform shape and size from adrenal and pancreatic tissue [17]. Another method, known as air–liquid interface, where cultures are maintained inside collagen gel in trans-wells [29], was successfully used to propagate PDOs from cancer cells and nonimmune as well as immune tumor stromal components from 100 patients representing 28 different types of tumors from various organs [30]. These PDOs could be passaged within one month at a success rate of 85% and cryopreserved. Other high-throughput methods have employed specialized seeding geometries of Matrigel-cell slurry as mini-rings in normal microwell plates [31] and bioprinting, where cells are deposited onto a support mesh and monitored by high-speed live-cell interferometry (HSLCI) [32].

Patient-derived organoids (PDO) can be established from both surgical samples as well as miniscule starting material from fine-needle biopsies of tumor patient samples [33]. PDOs in vitro can recapitulate tumor characteristics and its microenvironment both at the genetic and biological level [34,35,36]. Therefore, PDOs are considered avatars of the original tissue and can be used to move from bench-to-bedside to identify successful therapies to target primary and metastatic cancers in patients. Consistently, a few studies have demonstrated that the response of PDOs to chemotherapies positively correlated with the response of the patients to the same treatments [34,37,38]. Living biobanks of PDOs have been established from matched healthy and tumor tissues [34,39,40,41], where organoids are expanded in vitro and then frozen in liquid nitrogen and stored for future studies. These living biobanks are not only useful for screening of drug response or resistance and the discovery of biomarkers and biological targets, but they are also the gateway to personalized therapeutic approaches where tumor therapies are first evaluated in vitro and then adapted to the individual patient. In general, PDOs contain heterogeneous tumor cells, can be generated in the range 1–3 weeks and are relatively faster and more cost-effective compared to establishing patient-derived xenografts where tumor cells are transplanted into immunodeficient mice. As part of high-throughput screening platforms of hundreds of therapeutic agents, PDOs can provide results as fast as one week [31] to a few months, depending on the growth rate of the tumor [34,37].

In the following sections, we provide an overview of the recent approaches to generate 3D models in different endocrine tissues and the hormone-responsive organs, breast and prostate. These spheroid/organoid models are well suited to identify targets and new therapeutic options for cancer patients. Patient-derived organoids can additionally inform cancer therapy in real time.

## 3. Pituitary

The pituitary is a complex organ that synthesizes and releases peptide hormones to regulate basic physiological functions and consists of the adenohypophysis moiety (neuroendocrine hormone-secreting epithelial cells) and the neurohypophysis moiety (hypothalamic axon terminals and supporting stroma) [42]. This master gland regulates the function of other endocrine glands and endocrine target organs through the release of adrenocorticotropin, follicle-stimulating hormone (FSH), luteinizing hormone (LH), growth hormone (GH), prolactin, thyroid-stimulating hormone (TSH), oxytocin and vasopressin. The pituitary is highly plastic throughout life, constantly altering cell type ratios to keep pace with the demands and changes in physiology [43]. The mechanisms of pituitary regeneration are thought to involve resident adult stem cells. Accordingly, Cox et al. recently reported the first organoids generated from adult mouse pituitary, possibly from pituitary stem cells as they are characterized by the expression of Sox2 [44]. Organoids from normal and regenerating pituitaries formed dense and cystic organoids, respectively, and displayed differences in transcriptome and molecular properties, possibly reflecting the activation state of the stem cells. Other groups have focused on the development of components of the hypothalamic–pituitary axis through the in vitro differentiation of organoids from embryonic stems cells (ESC) and induced pluripotent stem cells (iPSC) [45,46].

Most pituitary tumors arise from the neuroendocrine epithelia, are well differentiated, mostly nonmetastatic and have been redefined recently as pituitary neuroendocrine tumors (PitNETs) [47,48]. Four immortalized cell lines originating from human pituitary adenomas exist to date: HPA [49], GX [50], HP75 [51] and PDFS [52]. Human primary cultures can be maintained as 2D cultures in vitro for a few passages and are mostly used to decipher the mechanisms of hormone secretion [53,54,55,56]. Although PitNET organoid literature is sparse, a few publications have emerged in the last few years. Nys et al. generated organoids in Matrigel from 21 human tumors with 95% success rate, although the organoids could not be expanded in vitro [57]. These organoids reflected the morphology and transcriptome of parental tumors, expressed stem cell markers, but were not tested functionally. Chakrabarti et al., on the other hand, successfully generated and expanded PitNET organoids in Matrigel from 35 tumors of patients with Cushing’s disease [58]. The organoids were composed of multiple cell types, including stem cells and PDOs from functional tumors successfully secreted ACTH in vitro [58]. In a high-throughput screening of 83 drugs, the PDOs from each patient displayed variable responses to the drugs, mirroring the genetic and molecular differences inherent to the individual patients [58]. Finally, Mallick et al. used iPSC and PDOs for mechanistic studies of new treatments alone or in combination with existing PitNET therapies [59].

## 4. Thyroid

Thyroid follicular cells, also called thyrocytes, are the major cell type in the thyroid. They produce the thyroid hormones triiodothyronine (T3) and thyroxine (T4), which are stored and secreted in a complex series of reactions [60]. Thyroid follicular cells are of epithelial origin, forming a monolayer between the colloid-filled lumen and the extrafollicular space. The thyroid also contains a second type of hormone-producing cells named parafollicular cells, or C cells, which are classified as neuroendocrine and primarily synthesize the hypothalamic hormone calcitonin. Additionally, the thyroid contains a network of capillaries surrounding each follicle, thus providing systemic delivery of the released hormones. The stromal compartment, which encapsulates the follicular thyroid tissue, consists mainly of neural crest-derived ectomesenchymal fibroblasts [61]. It also contains macrophages and mast cells (MCs) that are likely to be involved in thyroid cancer development and progression [62].

Thyroid cancer (TC) is the most common endocrine malignancy, accounting for 3.4% of all cancers diagnosed annually. It is classified into three main histological types: differentiated (papillary and follicular PTC/FTC), undifferentiated (poorly differentiated and anaplastic TC), and medullary TC, which arises from the neuroendocrine parafollicular cells [63]. Treatment of well-differentiated TC usually involves surgical resection of the thyroid gland, followed by treatment with radioactive iodine (I131) for total ablation [64]. Differentiated TC accounts for more than 90% of thyroid malignancies. Though showing favorable survival outcomes (5-year survival rates of 95–97%), around 20% of PTC patients exhibit tumor recurrence, metastasis and radioactive iodine-refractory disease (RAIRD) within 10 years. The majority of TCs display genetic changes related to their histological classification. While PTCs most commonly harbor point mutations in BRaf, mutations of Ras family members are predominantly found in FTC. Anaplastic thyroid cancer (ATC, the most aggressive form of TC) frequently displays point mutations in both BRaf and Ras, as well as in the genes TP53 and β-catenin [65].

Unfortunately, most TC cancer cell lines are profoundly dedifferentiated and have gene expression profiles similar to ATCs even though many of them originated from differentiated cancers. Many of them have additionally acquired telomerase reverse transcriptase (TERT) promoter and tumor protein p53 (TP53) mutations under cell culture conditions (e.g., 8305C, BCPAP, TCO1). Patient-derived TC-organoids might offer a possibility to overcome these problems; the few studies on TC organoids published so far suggest genomic stability, even during long-term passaging. For instance, Chen et al. created PTC organoids from patient tissue which recapitulated the histological characteristics, preserved their mutational landscapes and demonstrated patient-specific drug responses [66]. Drug screening further revealed that BRaf inhibitors, especially in combination with MEK inhibitors, RTK inhibitors, or chemotherapeutic agents were highly efficacious in BRAF V600E mutant but not in BRaf wild-type organoids. Sondorp et al. demonstrated that PDOs recapitulate PTC tissue characteristics and phenotype of radioactive iodine (RAI)-refractory disease (RAIRD), thus potentially enabling the early identification of I131-resistant patients [64]. While TC organoid research has focused on PTC so far, Pecce et al. recently established the first ATC-derived organoid [67].

However, efforts have been made not only to generate TC organoids for preclinical studies but also to use organoids from normal thyroid tissue for regenerative medicine purposes. Patients who underwent total bilateral thyroid resection require lifelong thyroid hormone replacement, and many of them suffer from therapy-induced symptoms that decrease their quality of life. Engineering functional thyroid tissue could provide a potential therapy for the many athyroid patients [8,9,10]. These models have been developed from murine models and provide proof of principle that functional thyroid tissue can be created in vitro. Antonica et al. was the first to develop a mouse ESC-based 3D thyroid model. While transient overexpression of Nkx2.1 and Pax8 directed the differentiation of mESCs into thyroid follicular cells, TSH supplementation induced their rearrangement into 3D follicular structures, which rescued thyroid hormone plasma levels when grafted into athyroid mice [68]. Based on this groundbreaking work, several groups succeeded in establishing functional thyroid follicles from human ESCs (hESCs) by overexpressing Nkx2.1 and Pax8 [69]. Kurmann et al. first identified bone morphogenetic protein (BMP) 4 and fibroblast growth factor (FGF) 2 as key regulators for thyroid lineage specification from endodermal cells [70]. By stimulating BMP and FGF signaling pathways, they were able to produce human thyroid progenitors from normal and disease-specific iPSCs generated from patients with hypothyroidism [70]. Since then, further steps toward thyroid replacement have been made. For instance, Bulanova et al. created a functional vascularized mouse thyroid gland constructed from embryonic tissue spheroids by 3D bioprinter technology [71]. Transplantation of these constructs under the kidney capsule of hypothyroid mice normalized blood thyroxine levels, thus demonstrating proof of concept [71]. Though the 3D models described above were rodent-derived and lacked a functional vascular network, these experiments are promising and establish the way for future studies with human samples.

## 5. Adrenal

The adrenal gland is formed by two components of distinct embryological origin: the cortex, involved in steroidogenesis (glucocorticoids, mineralocorticoids, androgens), and the medulla, involved in catecholamine production. The adrenal field has historically suffered from lack of appropriate in vitro models to study the complexities of adrenal function, hormone production and mechanisms of cancer progression [72]. Although primary human fetal and adult cortical and medullary cells are difficult to isolate, recently, Poli et al. reported the spontaneous organization of mixed fetal adrenal cells from both the cortex and medulla into organoids in normal tissue culture plates [73]. The arrangement and function of these 3D structures resembled that of the embryonic organ in vivo, classifying them as a new in vitro model to study the development of the human adrenal. In a complementary study, organotypic slices of human embryonic adrenals could be maintained in hanging drop configuration for mechanistic studies of steroidogenesis [74]. Human cell lines for the adrenal medulla do not exist to date and those for the adrenal cortex are discussed below [75,76].

Adrenocortical carcinoma (ACC), originating in the adrenal cortex, are rare tumors, often metastatic when diagnosed. Unfortunately, systemic therapies are mostly ineffective, leading to an overall 5-year survival rate of 22–44% depending on the staging and surgical options [76,77]. A panel of immortalized cell lines from ACC has been successfully generated, including NCI-H295 [78], MUC-1 [79], CU-ACC1 and CU-ACC2 [80], Jil-2266 [81] and TVBF-7 [82]. While NCI-H295 and JIL-2266 were derived from primary ACC prior to systemic treatment, CU-ACC1 and CU-ACC2 originated from chemonaive ACC metastasis, while TVBF-7 and MUC-1 were derived from local and distant ACC metastasis after administration of multi-chemotherapeutic therapies. NCI-H295 was the gold standard for many years for steroidogenesis studies [83,84], toxicology of endocrine disruptors [85,86,87] and preclinical evaluation of cancer therapies [88,89,90,91], and as early as 2012–2013, NCI-H295 spheroids were already used as an in vitro model of hyperaldosteronism [92] and to validate a high-throughput screen of 2816 clinically approved drugs [93]. We can already appreciate a shift in the past few years to spheroid/organoid cultures in adrenal cancer research. Krokker et al. found that spheroids formed in Matrigel and ULA secreted a higher concentration of steroids than monolayer cells, albeit the methods to generate Matrigel spheroids were technically challenging [94]. Silveira et al. identified a milder effect for mitotane in H295 spheroids than monolayer cells, unsurprising since spheroids are more resistant to therapies due to inhibition of apoptotic mechanisms, hypoxia and diffusion limits for drugs [15]. In another report, the incomplete effect of the drug sorafenib on apoptosis was exposed only in H295 spheroids and not monolayers [95]. The conditions created in H295 spheroids treated with sorafenib were permissive for the remaining drug-resistant cells to acquire an invasive phenotype. These results with sorafenib and H295 spheroids, but not monolayer cultures, echoed the increase in metastatic lesions in cancer patients of a failed clinical trial of sorafenib [96]. Recently, two interesting studies addressed the main challenges regarding the ACC standard-of-care drug mitotane by implementing NCI-H295R spheroids. The authors aimed to overcome the well-known poor bioavailability and unfavorable pharmacokinetics of this drug by the successful development of three potentially injectable nanoparticle formulations: a micellar [97], a liposomal and an albumin-bound variant [98]. The newly developed albumin-stabilized formulation not only displayed increased cytotoxic properties, but furthermore caused the disintegration of the spheroids, which was not detectable for the parental drug [98]. Interestingly, Avena et al. demonstrated that XCT790, an inverse agonist for estrogen receptor α, was able to reduce 3D spheroid formation and motility not only in H295R cells, but also in the mitotane-resistant MUC-1 model, suggesting that such an approach could be another effective option for the treatment of mitotane-resistant ACC [99]. Additionally, there is one report of PDOs established from fine-needle biopsies of human tumors. Baregamian et al. [100] reported the successful generation of organoids from ACC and adrenal neoplasm samples in ultra-low attachment (ULA) plates. These organoids were formed in the time range 1–3 weeks and maintained cortisol secretion for only 2–3 passages.

To further explore the feasibility of adrenal spheroid/organoid models in a high-throughput format, our laboratory initiated studies with the adrenal cell lines NCI-H295 and MUC-1, various benign, malignant medullary and cortical adrenal cancers along with primary adrenal organoid cultures from bovine/porcine tissues on Sphericalplate 5D [17]. In this study, spheroid formation for TVBF-7 was also confirmed for the first time. Originally implemented as primary culture ACC115m, these cells were found to be continuously passageable, subsequently characterized as adrenocortical cell line and at that point renamed to TVBF-7 [82]. Additionally, mixed spheroids of cortical and medullary primary cells from bovine/porcine tissue were generated to recapitulate the adrenal gland in a dish. Expression of appropriate markers and secretory profiles revealed that these mixed spheroids are healthy and functional.

In contrast to ACCs, pheochromocytomas (PCCs) and paragangliomas (PGLs), together referred as to PPGLs, arise from chromaffin cells of the adrenal medulla or from the sympathetic or parasympathetic paraganglia. About 10–15% of PCCs and 35–40% of PGLs are metastatic [101,102,103,104,105,106]. Around 30–35% of PPGL patients show germline mutations and another 30–40% of PPGLs bear somatic driver mutations [107,108,109,110,111,112], as recently reviewed [113]. Based on these genetic mutations and pathogenetic pathways, PPGLs are classified into three main clusters. Cluster 1 includes mutations in genes involved in pseudohypoxia signaling (genes of cluster 1A: *SDHx*, *FH*, *MDH2*, *IDH*, *GOT2*, *SLC25A11*, *DSLT*; genes of cluster 1B: *PHD1/PHD2*, *VHL*, *HIF2A*). Cluster 2 is characterized by alterations in kinase receptor signaling and protein translation pathways (genes of cluster 2: *RET*, *NF1*, *TMEM127*, *KIF1B*, *MAX*, *FGFR1*, *MET*, *MERTK*), while cluster 3 is Wnt-signaling-related, reviewed in [113]. PPGLs typically exhibit low growth rates, with estimated doubling times of 4 to 7 years, rendering it difficult to establish cell-line models for the adrenal medulla. However, while human PPGL cell lines are still lacking, culturing pheochromocytomas from animal tumors led to the development of the rat PC12 (carrying a *MAX* gene variant) [114] and RS0 cell lines (lacking *SDHB*) [115] and the MPC [116] and MTT [117] cell lines, originating from *NF1* knockout mice. Moreover, Bayley et al. successfully cultured human adrenal PCCs and extra-adrenal PGLs over long periods. While a small number of synaptophysin/tyrosine hydroxylase-positive chromaffin cells persisted for up to 99 weeks, unfortunately a viable PPGL cell line could not be established [118]. The Nölting group has established human PPGL primary cultures from surgery-derived patient tumors for multiple drug testing in individual patient tumors and correlated their results with the germline or somatic driver mutations of the respective tumors [119,120]. They cross-validated their results in spheroids from MPC/MTT cell lines (including *SDHB* knock down spheroids) and in 3D human PPGL primary culture models [119,120]. Finally, Calucho et al. [121] reported at the 2023 AACR conference that they generated 10 organoids from patient paragangliomas with conserved secretory profile and drug sensitivity of the original tumor. Altogether, these efforts may establish the way for personalized therapy approaches in PPGLs.

## 6. Pancreas

The pancreas is an organ with both exocrine and endocrine moieties. The exocrine tissue makes up more than 95% of the pancreas while less than 1–2% of the tissue has endocrine function [122]. The exocrine acinar cells secrete into the exocrine ductal system more than 20 different enzymes necessary for the digestion of food, which are then released into the duodenum [123]. The endocrine pancreas constitutes of the Islets of Langerhans , composed of numerous cells types, which produce hormones that regulate whole-body energy homeostasis and exocrine pancreas function. The α–β–δ–ε– and PP/F cells produce glucagon, insulin, somatostatin and ghrelin pancreatic polypeptide, respectively [124]. Islets, acinar cells and various other cells types are held together by the ECM, which provides structural integrity to the pancreas as well as signaling cues due to its biochemical composition and concentration of signaling molecules such as growth factors and cytokines [125]. Although functional and biochemical studies of acinar cells and islets are preferentially conducted with primary cells, these cell types are difficult to culture since they dedifferentiate/transdifferentiate after a few days in culture (acinar) or survive for up to one week post isolation (islets). In vitro models for both cell type are restricted to a few rarely used pancreatic cancer lines for acinar cells, while for beta-cell and diabetes research there are rodent insulinoma cell lines such as MIN6 (mouse) and INS-1 (rat) [126]. Recently, the EndoC-bH5 cell line, originating from human fetal beta-cells, was developed and displays similar insulin functional response to adult human islets [127]. Interestingly, EndoC-bH5 can form functional spheroids in vitro.

Initial efforts to generate in vitro spheroids from endocrine pancreas were linked to human islet transplantation research, where survival of transplanted islets in diabetic patients was severely compromised by hypoxia and limited vascularization. Parallels can be drawn between large organoids and large islets, where both suffer from inadequate diffusion of oxygen and nutrients and both can harbor necrotic cores. Accordingly, since survival and performance of smaller islets supersedes that of larger ones in vitro and following transplantation [128], many groups have focused on isolating islets, dissociating them into single cells, and then allowing their reaggregation as pseudoislet spheroids. Multiple reports have demonstrated that human pseudoislets contain more insulin, perform better than native islets in glucose-stimulated insulin secretion (GSIS) in vitro and improve hyperglycemia in vivo when transplanted in mice [6,7,17]. Regarding efforts to improve vascularization, Wassmer et al. reported that mixed organoids consisting of islet cells, HUVAC and hAEC cells outperform when transplanted in mice and display adequate vascularization [129]. Therefore, re-engineering islet size and cell type might positively affect survival of pseudoislets and the success of human transplantations. Another source of islets closely resembling the biology of human islets come from the neonatal pig, where pseudoislets generated in Sphericalplates 5D display superior insulin response in vitro and in vivo [130].

Most advances in PDOs and pancreatic cancers have taken place with pancreatic ductal adenocarcinomas (PDAC), where living organoids biobanks from the original cancer, the surrounding healthy tissue or metastasis have been established by multiple groups with a 70–85% success rate [131,132,133,134,135]. PDAC, a devastating disease arising from exocrine pancreas with a 5-year survival rate of <9%, represents 95% of all pancreatic cancers and is diagnosed mostly at an advanced stage due to lack of clinical symptoms and biomarkers [136]. In the last few years, the use of organoids in PDAC preclinical research has rapidly progressed, where researchers have mostly focused on optimizing growth conditions and characterization of the PDOs in vitro to evaluate their applicability as patient avatars [131,133,135,137,138]. Overall, PDAC organoids can be successfully generated from surgical resection and fine-needle biopsy samples, and reflect the genomic, molecular and architectural profile of the original tumors [37,134,139,140]. The response of PDOs to therapies, such as gemcitabine, is variable and cannot always be compared to patient response due to the rapid progression of the disease in humans [134]. However, two reports of high-throughput screens using PDOs favor efforts in personalized medicine since PDOs captured the inter- and intrapatient variability in response to drugs (single or in combination) and reflected patient sensitivity to treatment [34,41,141]. The time needed to generate enough PDOs for such high-throughput screens, however, is on the order of 2–3 months.

Two to five percent of pancreatic cancers are pancreatic neuroendocrine neoplasias (pNEN), a heterogeneous group of epithelial tumors with very rare occurrence (<1 in 100,000) [142]. They arise from the various cells types of islets and can be classified into functional (secretion of hormones: insulinomas, glucagonomas, somatostatinomas, etc.) and nonfunctional (no hormones secreted, most common form). At the genetic level, the four main pathways consistently altered in pNENs are PI3K/mTOR signaling, chromatin remodeling, telomere alterations and DNA repair [143]. Treatment options for metastatic pNEN have poor response rates and include somatostatin analogue (SSA), everolimus, sunitinib, temozolomide, streptozocin and peptide receptor radionuclide therapy with radiolabeled SSA. For localized tumors, however, surgery is the best therapy to date [144]. Consequent to the lack of early diagnostic biomarkers, pNENs are discovered at late stages and associated with a survival rate of 4–5 years. Common markers for pNENs include chromogranin (CgA) and synaptophysin. pNETs expressing the somatostatin receptors SSTR2/SSTR5 are the most frequent and associated with better survival, since SSTR2 is the main receptor for somatostatin analogue treatments (octreotide and lanreotide) and radiotherapy. Cell lines such as BON-1, QGP-1 and NT-3 are widely used in pNEN research. Compared to 2D culture, spheroids from BON-1 and NT-3 display resistance or require higher concentrations of certain drugs to elicit effects on cellular viability and apoptosis [145]. These attributes of spheroids versus monolayer cultures are thought to arise from diffusion kinetics and hypoxia within the 3D aggregates, as mentioned above. Although BON-1 spheroids can be generated in agarose-coated plates [146], Bresciani et al. [27] showed that BON-1 spheroids generated in ULA plates are the most suitable regarding ease of use and reproducibility of results, underscoring again the need for high-throughput methods when working with spheroids. Our own experiments further indicate that BON-1 can be effectively cultured on Sphericalplates 5D, thereby representing an additional option to generate BON-1 spheroids (unpublished data). Primary pNET cell cultures with neuroendocrine characteristics have been established by a few groups and used to investigate intracellular mechanisms of drugs and their targets [147,148,149]. Mohamed et al. [148,149] explored the effect of everolimus (mTOR inhibitor) alone and in combination with somatostatin analogues (SSA) octreotide and pasireotide on primary cultures of pNET. Furthermore, the therapies suppressed cell proliferation and CgA secretion in these pNET primary cultures similarly to in vivo, which was not always effective in BON1 cell lines [150]. PDOs, on the other hand, are difficult to establish. To date, the only pNEN organoid line was reported by Kawasaki et al. and was one out of twenty-five NEN organoid lines established by the group [151]. Short-lived 3D tumoroids from human primary pNETs have been reported and used to investigate response to current treatments (such as sunitinib, everolimus and temezolomide) and identify novel therapies targeted against PI3K and CDK4/6 signaling pathways [145,152].

## 7. Ovaries

The ovaries are endocrine glands responsible for the production of estrogen and progesterone and the regulation of female reproduction. They contain a lifetime supply of oocytes which, starting at puberty, grow and develop monthly in the follicle under the influence of follicle-stimulating hormone (FSH) released from the pituitary. The follicle, consisting of the oocyte surrounded by granulosa and theca cells, undergoes folliculogenesis during ovulation marked by growth, proliferation, differentiation and restructuring of the various cell types and the ECM [153]. To better understand this process, and to preserve fertility in humans and other species, in vitro follicle culture has been established for various organisms. Follicles are isolated by mechanical or enzymatic digestion and maintained as 3D organoids to successfully preserve early stages of follicle development and long-term culture in vitro [154]. Methods for 3D cultures include suspension culture (similar to hanging drop but in 96-well format [155]), encapsulated culture (where follicles are embedded in collagen, agar, alginate beads [156]) or a multistep culture (combination of these methods), which was recently used to achieve complete human oocyte development in vitro from isolated primordial follicles [157]. Additionally, primary human cultures from healthy fallopian tubes reflect the in vivo secretome of the tissue [158], and organoids can be generated from stem cells, are stable and can be expanded for longer than one year in vitro [159].

Ovarian cancers are mostly detected at advanced stages and are the most lethal gynecological cancers in women with a 5-year survival rate of 47% [160]. Treatments include surgical resection when possible accompanied by carboplatin and paclitaxel chemotherapies [161]. Although initial response to chemotherapy treatment is favorable, lesions soon become drug-resistant and patients relapse [162]. Ovarian cancers are a heterogeneous group of cancers that share an anatomical location and originate mostly from nonovarian tissue, such as the fallopian tubes, the ovarian surface epithelium (OSE), which surrounds the ovaries, the endometrium and other distal sites such as gastrointestinal tumors [162]. These cancers fall under three main categories, borderline tumors (noncarcinomas) and type I and type II tumors (carcinomas); 70–80% of mortalities arise from type II high-grade serous (HGS) carcinomas, the most common form of ovarian cancer thought to originate from fallopian tubes or OSE [163].

Multiple cell lines are available to explore the mechanisms of ovarian cancer in vitro and many groups have assessed their usability as model systems [164,165,166,167,168]. Reports using these cell lines as spheroids have focused on determining differences between monolayers and spheroids. Nowacka et al. found that spheroids are more resistant to cisplatin and paclitaxel than monolayer cultures [169], while Heredia-Soto et al. reported that 16 immortalized cell lines grown as spheroids displayed increased resistance to cisplatin and increased expression of markers of epithelial–mesenchymal transition compared to monolayers [16]. Interestingly, Casagrande et al. determined that factors released from activated platelets induce spheroid formation in ovarian cancer cell lines and contribute to drug resistance [170]. Furthermore, Sodek et al. established a positive correlation between spheroids and increased invasiveness of immortalized cells [171]. To evaluate characteristics of invasiveness and chemotherapy resistance, Novak et al. applied compressive stimuli in a bioreactor to cell lines encapsulated in a special ECM made from agarose and collagen [172]. It would be interesting to determine in future studies if compression in a bioreactor would increase as well the invasiveness and drug resistance of spheroids or PDOs.

Work with organoids is progressing rapidly, where biobanks from primary tumors and their metastasis from multiple ovarian cancer subtypes have been established [163,173,174,175]. Whole-genome sequencing and histology assessments indicate that PDOs from ovarian cancers mirror the features of parental tumors in addition to sharing similarities in response to chemotherapy treatments [38,163,176,177,178]. PDOs can be established from multiple ovarian cancer subtypes, including nonmalignant samples, with a success rate of 65–90% [163,174,179]. Interestingly, resistance to cisplating was linked to glucose-6-phosphate dehydrogenase (G6PD) and glutathione-producing redox enzymes in PDOs from different types of ovarian cancers [180]. PDOs from HGS can be generated in less than one week [181] and were used to evaluate the origins and progression of this tumor subtype [182] and the state of the DNA repair machinery [176]. Furthermore, there are a few reports that support the feasibility of conducting drug screens for therapies with PDOs from ovarian cancers. In the fastest turnaround time of 10 days, de Witte et al. demonstrated congruence with intra- and interpatient heterogeneity in response to six treatments in PDOs from 23 patients [38]. Jabs et al. screened 22 drug therapies (single and in combination) and correlated their effect to genomic alterations in organoids and 2D culture using DeathPro, an automated confocal microscopy system that generates drug efficacy values over time [177]. Finally, using the mini-ring seeding method, Phan et al. obtained results from a high-throughput screen of 240 kinase inhibitors in PDOs during only one week’s time [31]. The authors could generate six 96-well plates per patient and test each drug at two different concentrations, for a total of 480 conditions, and identified one compound that could effectively target all tumors.

## 8. Breast

Mammary gland development is mainly controlled by the ovarian steroid hormones, estrogen and progesterone, and the pituitary hormones, growth hormone (GH) and prolactin. Unlike other tissues, the mammary gland undergoes its major developmental processes after birth, particularly during pubertal development and pregnancy. At puberty, a highly branched ductal epithelium emerges from a rudimentary embryonic ductal tree to infiltrate the stroma. Under combined action of progesterone and prolactin during pregnancy, milk-producing lobuloalveolar structures (alveoli) differentiate from the terminal ductal units and are connected to the nipples [183]. After parturition, the alveoli further expand and take up the space of regressing mammary stromal adipocytes, thereby multiplying epithelial volume many times [184]. During involution, epithelial volume is drastically reduced, a process which includes programmed cell death of the epithelium, remodeling of the extracellular matrix (ECM) and redifferentiation of adipocytes [184,185,186]. Morphologically, mammary glands are formed by several different types of cells. The branching ductal epithelium comprises three different epithelial lineages. The hollow lumen is surrounded by two populations of luminal cells (LEP), which are responsible for milk production and hormone sensing. In contrast, the basal epithelium consisting of contractile myoepithelial cells (MEP) form the outer layer of mature mammary ducts that serve in milk ejection during breastfeeding. It also harbors adult mammary stem cells (MaSCs), capable of reconstituting a complete mammary epithelial ductal structure when implanted as single cells into a cleared fat pad in vivo [187,188]. The epithelium is surrounded by a basement membrane and embedded in a complex stroma containing fibroblasts, adipocytes, macrophages, nerves, vasculature and lymphatics [183,187]. As mentioned before, morphogenetic and functional changes in the mammary gland, particularly during pubertal development and pregnancy, are highly dependent on hormones but also on signals from the surrounding nonepithelial cells and the ECM.

Three dimensional cell culture models to study regulatory mechanisms of breast cell behavior have been widely used for decades. Among the 3D models, primary mammary organoids have played a major role. They are commonly established from epithelial fragments, sourced either from mice or from human tissue obtained by reduction mammoplasty, prophylactic mastectomy, breast biopsy or resected cancers. The main challenges of long-term organoid culture remain the propagation of the multiple mammary epithelial cell types (especially progenitor cells) and the maintenance of hormone receptor expression. New techniques have recently been developed to preserve all major mammary epithelial lineages and to retain characteristic expression patterns of mammary markers over long periods [189,190]. While early mammary organoid models were mostly morphologically simple structures that did not resemble the morphology and function of the in vivo mammary gland, efforts have been made to increase the complexity and functionality. Dynamic shape changes in response to growth factors and ECM cues have been demonstrated in a number of studies [191,192]. Very recently, Caruso et al. [193] obtained a correctly branched structure by alternating the addition of fibroblast growth factor 2 and epidermal growth factor, while Sumball et al. [194] recently succeeded in generating functional primary mammary organoids mimicking lactation. Upon prolactin stimulation, these organoids produced milk for at least 14 days, thereby maintaining their histologically normal bilayered architecture.

Breast cancers (BC) typically originate in the luminal ducts. They progress through a series of defined structural changes, beginning with the filling of the lumen, invasion of the cancerous cells past MEP and finally to metastasis. Most ductal carcinomas in situ (DICSs) remain limited to the epithelium, suggesting that MEP represents a physical barrier to invasion [183,195]. BC is the most frequently diagnosed cancer and most common cause of cancer death among women worldwide [196]. BCs present themselves as a heterogeneous tumor family, usually categorized by their hormone receptor status for estrogen receptor (ER), progesterone receptor (PR) and human epidermal growth factor receptor 2 (HER2). Luminal-A-like tumors are ER+/PR+/HER2−, luminal-B-like tumors are ER+/PR−/HER2− and HER2-enriched ER+/PR−/HER2+, while tumors lacking ER, PR and HER2 expression are referred to as triple-negative and are considered to be the most aggressive. BC also comprises a rare group of neuroendocrine neoplasms classified as “neuroendocrine tumor of the breast (Br-NET)”, “neuroendocrine carcinoma of the breast (Br-NEC)” and “mixed neuroendocrine/non-neuroendocrine neoplasm of the breast (Br-MINEN)”, accounting for less than 1% of all neuroendocrine tumors and an overall incidence among breast cancer between 0.1% to 5% [197].

The most common cell lines to mimic luminal A BCs (ER+/PR+/HER2−) are MCF-7 and T-47D, while ZR75.1 and BT474 cell lines are widely used to represent luminal B BCs. UACC-893 and SKBR3 cells are cell lines derived from HER2+ BC [195,198]. Unfortunately, no neuroendocrine BC cell lines have been described to date. Different protocols have been used to generate spheroids from these immortalized lines with varying degrees of success. While MCF-7 cells seem to form spheroids under all conditions, SKBR3 spheroid assembly was not inducible under any condition, while MDA-MB-231 formed spheroids only with the liquid overlay method [199]. As cancer-associated fibroblasts (CAFs) constitute up to 80% of the BC mass, many coculture studies with different immortalized cells and various fibroblasts have been performed. Coculture with fibroblasts significantly improved the ability of BC cell lines to form spheroids [200]. Furthermore, invasive properties and drug resistance were increased compared to homogeneous BC spheroids [195,201,202,203]. Kaur et al. showed that spheroids from UACC-893, BT20 or MDA-MB-453 cultured with foreskin fibroblasts showed the immunocytochemical characteristics of BCs [204]. Recently, several coculture models employing MDA-MB-231 cells and fibroblasts in combination with HUVECs and/or osteocytes have been successfully established using complex microfluidic systems and 3D bioprinting [195,205,206,207]. Furthermore, in order to mimic BC cell heterogeneity, spheroids consisting of BC cells with high (BT20 cells), low (MCF-7 cells) and no (MCF-10A cells) invasive potential have been created [208], However, the seeding ratio of such coculture models seems to significantly affect spheroid quality. In the study of Tevis et al. MDA-MB-231 spheroid formation was optimized by addition of macrophages in a 2:1 ratio [209]. Likewise, exceeding the seeding ratio of 2:1 in a MCF-7:MRC-5 (fibroblast cell line) coculture model led to fibroblast clustering and lacking integration [203].

In contrast to spheroids, organoids, generated from patient tumors, can be obtained from different tumor stages. Owing to the self-renewal capacity of containing cancer stem cells (CSCs), they capture and retain histological and genetic BC heterogeneity, thus allowing physiologically relevant in vitro drug screens. In recent years, various protocols for the generation for PDOs have been developed. Those mainly include coculture with CAFs and/or immune cells (e.g., macrophages) but there are also protocols just adding niche factors like the Wnt-agonist R-spondin, TGF-β-inhibitor Noggin and EGF to the culture medium [190,195,210]. Furthermore, protocols for genetic manipulation and xenotransplantation of PDOs have been established, facilitating the identification and evaluation of potential targets for BC treatment [211,212]. To date, PDOs from breast cancers have been mainly been used for genomic and biological studies. For instance, Davaadelger et al. could demonstrate that BRCA1 mutations influence hormone response, especially for progesterone [213]. First studies comparing the sensitivity of patient-derived BC-organoids to the clinical treatment response found high correspondences [214]. However, the future of BC organoids has just begun. Initial biobanks have been established, providing a powerful tool for future personalized treatment approaches [189,190,195].

## 9. Prostate

The prostate belongs to the male reproductive system, is under the control of male sex hormones and androgen receptor (AR) signaling and consists of epithelium and stromal components. The epithelium is composed of luminal secretory cells (fully differentiated and expressing high levels of AR), basal proliferative cells (relatively undifferentiated with low AR expression) and a rare population of neuroendocrine cells [215]. The stroma, a fibromuscular mysenchyme, regulates the development, differentiation and function of epithelial cells and is also under androgen control [215].

First efforts to generate spheroids from normal epithelial prostate cells were published in 2001, where spheroids displayed luminal phenotype and retained functional AR signaling [216]. Subsequently, spheroids from predominantly prostate basal epithelia were shown to possess self-renewing properties, formed ductal-like structures and could be passaged and cryopreserved [217]. Although lumen cells are difficult to culture, Chua et al. were able to produce and passage organoids from benign human prostate epithelial cells for three weeks, containing lumen, basal and progenitor cells [218]. The Clevers group extended these findings and demonstrated that stem cells can arise from both basal and luminal epithelia by expanding organoids from these two cell types in vitro [219]. These organoids originated from 40 independent human epithelial prostate biopsies and displayed stable genomic and molecular phenotype for more than 12 months [219,220]. Finally, in an effort to recreate the prostate in a dish, Richards et al. cocultured human epithelial prostate organoids with their stroma components [221]. Prostate stroma, but not nonprostate stroma, triggered multibranching of the organoids and increased both the efficiency of formation and the complexity of the branching.

Prostate cancer (PC) was the most diagnosed cancer in men in the US in 2022 [136]. Both androgen signaling and the prostate stroma play a critical role in the development and progression of PC [222]. Androgen signaling drives prostate cancer, and androgen deprivation therapies along with castration are standard of care [223,224]. Over the course of treatment, 20–35% of PCs develop resistance to androgen receptor inhibitors, known as castration-resistant prostate cancers (CRPC) [225,226,227]. A small percentage of CRPC then transition to the aggressive small cell neuroendocrine carcinomas, referred to as neuroendocrine prostate cancer (NEPC), which are treated unsuccessfully with platinum-based therapies [228,229]. Aside from androgen signaling, the stroma strongly influences TME and cancer growth by supporting abnormal ECM production and remodeling [222]. Prostate cancer is slow growing in vivo and in vitro, rendering it difficult to establish effective preclinical models to develop therapeutics [230]. Most current models are derived from advanced metastatic disease.

Immortalized cell lines heavily used in prostate cancer research include LNCaP [231], PC3 [232], DU145 [233], VCaP [234], MDA PCa 2a and 2b [235] and NCI-H660 [236]. As early as 1997, LNCaP spheroids successfully retained androgen signaling and secreted prostate-specific androgen only when cocultured with prostate stroma cells [237]. The group also investigated LNCaP spheroids cocultured with bone cells (on Earth and in the space shuttle Columbia) since PC metastasis often target bone and are supported by its microenvironment [238]. The Doran group used immortalized cell lines to investigate high-throughput methods to generate spheroids in normal tissue culture plates, first by engineering microwell inserts, and then by an improved system using microwell mesh to form 150 spheroids per well [239,240]. In both reports, spheroids were metabolically active, grew slower and were more resistant to docetaxel treatment than monolayers. In another study, culturing cell-line spheroids with cancer associate fibroblasts (CAF) decreased the response of the spheroids to antiandrogen therapies by enhancing their steroid and cholesterol biosynthesis [241]. Expression of relevant markers was enhanced in malignant versus benign tumor samples as determined by immunohistochemistry. Aside from spheroids of immortalized cell lines, other groups have focused on generating organoids from the LuCaP-PDX mouse models, a large group of advanced human pancreatic cancers in PDX form with vast genetic and molecular heterogeneity [242,243]. Although the xenograft cells were notoriously difficult to culture, Beshiri et al. were able to successfully produce organoids in Matrigel and demonstrated congruity with original PDX tumors by whole-exome sequencing [244]. Fong et al. cocultured cells from PDX models as organoids with osteoclasts embedded in hyaluronan gel and validated that osteoclast-PDX organoids reflected the in vivo architecture, cell–cell interactions and singling events of metastatic PC to bone [245]. Finally, the MURAL library of 59 PDX models was recently created, where organoids could be generated from some of the models and maintained for a few passages in vitro [246].

Organoids from human prostate cancer are difficult to generate, with success rates of 15–20%. Nonetheless, in a landmark study, Gao et al. described the first organoid biobank originating from six prostate cancer biopsies and one sample from circulating tumor cells with a success rate of 20% [247]. The authors demonstrated for the first time that organoids capture the diversity of solid tumors at the genetic and molecular level and displayed similar histopathology. In another study, four PDO lines from three needle core biopsies were successfully established and maintained for 10 weeks in vitro [244]. Organoids from NEPC have also been generated at a success rate of 16% and were nonetheless used in a screen of 129 therapeutic agents to find personalized solutions against this deadly subtype of PC [248]. Another study focused on understanding the effect of the ECM used to support the growth of organoids from one CRPC sample and one NEPC sample [249]. Tumor ECM was analyzed by transcriptomics, proteomics and immunohistochemistry and informed the composition of multiple variants of synthetic hydrogels. Hydrogel type and stiffness influenced the growth, epigenetics and gene expression in the organoids. Organoids were also used in a clinical trial to determine the efficacy of the drug alisertib, an inhibitor to Aurora kinase A, which is overexpressed in NEPC patients [228]. PDOs developed from two patients, one high-responder and one nonresponder, had the same response to the drug as the metastasis in vivo. Additionally, Pamarthy et al. reports that more clinical trials are using PDOs to evaluate their use in personalized medicine [250]. Finally, Servant et al. established new culture conditions for organoids originating from 81 samples of primary and metastatic prostate cancer, where 69% of organoids could be maintained long-term [251].

## 10. Future Directions

Novel therapies are direly needed to combat endocrine cancers. Three dimensional cultures are the appropriate models to identify drug targets and treatments, especially patient-derived organoids since they capture the molecular and genetic heterogeneity of in vivo tumors. We envision that, by culturing tumor biopsies as spheroids/organoids, the endocrine field can circumvent the difficulties associated with establishing new immortalized cell lines or working with animal models and advance not only basic research but cancer research and personalized medicine as well. PDOs offer the opportunity to identify successful therapies in vitro from patient tumors, and then successfully treat the patient with the same therapies. However, additional effort is necessary to understand appropriate culture conditions, to generate vast numbers of organoids in a short timeframe and to establish living organoid biobanks for many tumor types. Most of the cancers discussed here are identified when metastatic at advanced stages and with patients typically succumbing to the metastasis. 3D in vitro models offer the opportunity to tailor drug therapies to the original tumor and to heterogeneous metastatic lesions, thus identifying treatment that works in both instances [36].

## 11. Conclusions

Spheroids and organoids are 3-dimentional aggregations of cells in vitro which capture organ physiology and tumor characteristics at the functional and genetic level. Here, we have reviewed evidence that spheroids/organoids can be used to answer basic question in organ development and function as well as in preclinical cancer research. Spheroids/organoids can be potentially used as organ replacement or for drug screening and personalized medicine in cancer research. Although this technology is still in its early stages, in the near future spheroids/organoids can prospectively identify novel therapies to combat endocrine disorders and cancers. 

## Figures and Tables

**Figure 1 cancers-15-04601-f001:**
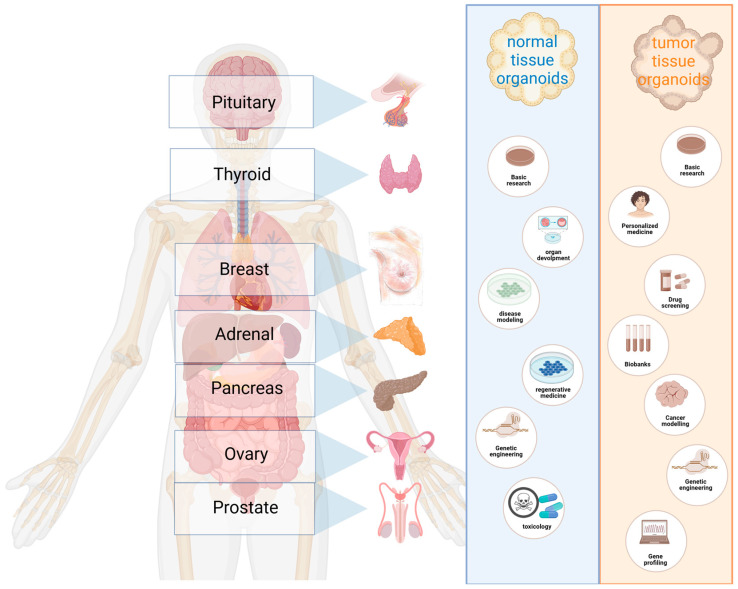
Applications for organoids from normal and cancerous tissue from the various endocrine glands discussed in this review. Created with BioRender.com.

**Figure 2 cancers-15-04601-f002:**
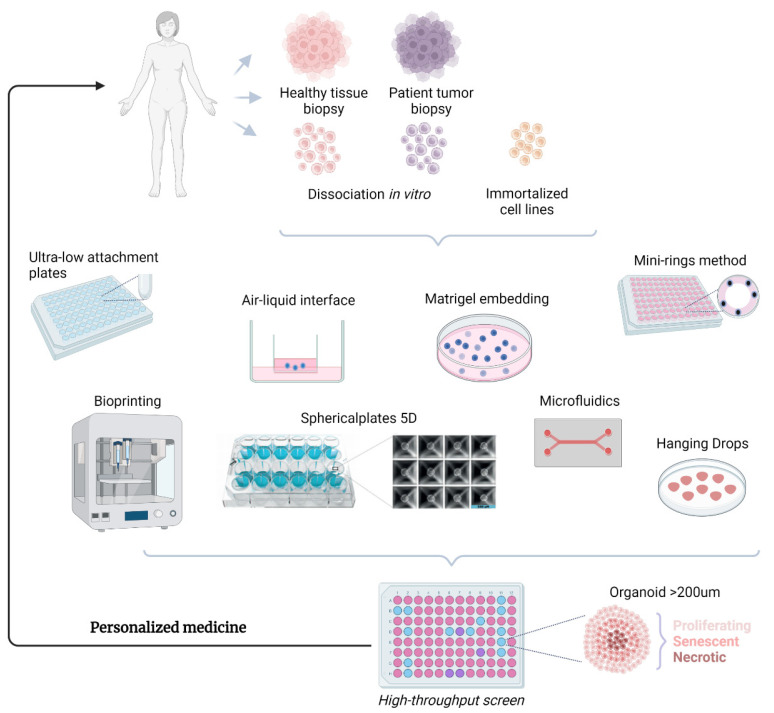
Generating organoids from patient biopsies of healthy or cancerous tissue, or spheroids from immortalized cell lines. Biopsies are dissociated and organoids are generated using the illustrated methods. They are then collected and used for high-throughput screening of individual therapies for each patient. Created with BioRender.com.

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
