# Peer review of "Three Dimensional Models of Endocrine Organs and Target Tissues Regulated by the Endocrine System"

_cancers, 2023, doi:10.3390/cancers15184601_

Round 1

Reviewer 1 Report

The present review delves into the development of organoids/spheroids of different tissues with endocrine component. The bibliographic review is good, performing a large bibliographic search and using recent citations. However, it has some errors: 

- In Figure 1 it should be made clear in the legend what each acronym corresponds to (GEP-NET, .....).

- The introduction should better reflect the connection between organoid development and the endocrine system. 

- The authors should emphasize the importance of the choice of culture medium in the development and expansion of organoids/spheroids. 

- Line 113; Rewrite the sentence as integrins and cadherins are not considered receptors. 

- Section 2 should better clarify what the disadvantages of organoid use are.

- Line 150: Rewrite the sentence

- Do organoids from PitNET recapitulate hormone release as in in vivo systems?

- Line 202: Rewrite the sentence

Paragraph 221-228: Need to introduce some reference 

Line 223: Add a dot after et al.

- The introduction made of the pancreas is not clear, it should be rewritten in accordance with the other introductions talking about the generalities of the organ in question. 

line 371: One space is missing

- Paragraph 463-465: Delete this sentence since it does not talk about the formation of spheroids.

- Line 535: What do you mean by "altering": do they change these growth factors for others? which ones?

- The conclusion is not supported in the review heha. The authors talk about how endocrine tissue organoids are formed. The conclusion could be rewritten

Reviewer 2 Report

Figure 1 is highly non-informative and includes data that are yet irrelevant to the text and could/should be omitted.

Reference 1 – either provide data (not quoting review) or omit the sentence – line 40.

Line 40 – result -> results

Chapter 2 almost completely neglects spheroids – I would add a paragraph dedicated to spheroids, mentioning the impact of 3D structure on spheroids (either single cell line or mixed cells) vs. 2D cell line. This discussion was mentioned in regard to PDOs that are much more complex considering their variable cellular consistency.

The last paragraph of section 2 is somewhat – self-referencing. The experience of the authors is important but should be added as part of a review of other methods and not only as a single example. Nevertheless, if the authors want to share their data, I would suggest adding detailed protocol, figures, and images as supplementary.

Section 3:

1.       Please check the phrasing in line 161

2.       I would suggest adding a table summarizing each of the papers described, providing more data on the method for each organoid and its parameters. These should probably include a number of passages, possible experiments, histological staining, and proliferative parameters (KI67, etc).

Section 4:

1.       The first sentence is not necessary, in my view. This is a nice attempt to connect sections but seems unnecessary.

2.       Calcitonin may have a role in calcium regulation, but its current role in calcium regulation is negligible.

3.      Line 201 – ATC?

4.    Line 208 – "Patient-derived TC-organoids might offer a possibility to overcome…" – How? Re-differentiation and disappearance of mutations? This is unclear.

5.      Line 215 – BRAF V600E – Please write gene names and variants correctly.

6.      Last paragraph of section 4 deals with generating normal thyroid tissue. It should be stated that hormone replacement after thyroidectomy is usually the easiest task in endocrinology. Thus, the interest in this technique might be from the perspective of modeling tissue from progenitor cells. If the authors find this issue relevant, I would suggest discussing this from the aspect of potentially inducing mutations typical for well-differentiated TC, enabling the development model of WD TC. Normal thyroid generation is not within the scope of this review.

Section 6:

Line 399-400 – The third genetic group is unclear to me – may this be –an undetermined group and not alpha more than beta? Please see several works by the Scarpa team, and by Thirlwell and coauthors (both according to methylation), and by Chan et al from Nat Comm. The reference (136) is not familiar to me in this regard.

In general, I would suggest looking at the work by Talya Dayton on NET organoids and by Hans Clevers. Even if not published as papers, they published numerous preliminary studies at conferences and are the leading in this field.

Section 8:

Again, the 1st two paragraph deal only with normal mammary gland structure and models. Even when dealing "only" with cancer models, this is an EXTREMELY wide topic (endocrine glands and target organs). Adding to this topic, normal tissue models seem too much for one review.

Section 9:

 PC is the most ?? diagnosed.

General comment:

It is very clear that the authors are experts in 3D models of endocrine cancer. I would urge them to list the "must do" and "must have" characteristics, previous pitfalls, and technical guidelines for future projects. This will probably make the review highly attractive for any scientist/group in the field.

NA

Reviewer 3 Report

Dear Author,

This review article needs to address few additional comments. Please follow the instruction from the editor.

---------------------------------------------------------------------------------

The review article "Three-dimensional models of endocrine organs and target tissues regulated by the endocrine system" by Edrila Luca et al focused on recent efforts in basic and preclinical cancer research on establishing methods to create organoids/spheroids and living bio banks to produce endocrine tissues and target organs under endocrine control,striving to personalized medicine solutions. Overall, I recommend this review article must meet the major comments before considering for the next level.   Major comments:
1. This review article did not explain the clinical aspects and future translational approach.  2. I recommend figures for each endocrine system to explain the graphical organoid functions.  3. Recent advances and update in personalized Medicine and clinical trials.

-----------------------------------------------------------------------------------------------------------------------------

Thanks.
